# Deciphering the Path of *S-nitrosation* of Human Thioredoxin: Evidence of an Internal NO Transfer and Implication for the Cellular Responses to NO

**DOI:** 10.3390/antiox11071236

**Published:** 2022-06-24

**Authors:** Vitor S. Almeida, Lara L. Miller, João P. G. Delia, Augusto V. Magalhães, Icaro P. Caruso, Anwar Iqbal, Fabio C. L. Almeida

**Affiliations:** 1Institute of Medical Biochemistry Leopoldo de Meis (IBqM), Federal University of Rio de Janeiro (UFRJ), Rio de Janeiro 21941-590, Brazil; vitor.almeida@bioqmed.ufrj.br (V.S.A.); l.miller@nano.ufrj.br (L.L.M.); gama@nano.ufrj.br (J.P.G.D.); avieira@bioqmed.ufrj.br (A.V.M.); icaro.caruso@unesp.br (I.P.C.); a.iqbal.chem@gmail.com (A.I.); 2National Center for Structural Biology and Bioimaging (CENABIO), Federal University of Rio de Janeiro (UFRJ), Rio de Janeiro 21941-590, Brazil; 3Institute of Chemistry, Rural Federal University of Rio de Janeiro (UFRRJ), Seropédica 23897-000, Brazil; 4Multiuser Center for Biomolecular Innovation (CMIB), Department of Physics, Institute of Biosciences, Letters and Exact Sciences (IBILCE), São Paulo State University (UNESP), São José do Rio Preto 15054-000, Brazil; 5Department of Chemical Sciences, University of Lakki Marwat, Lakki Marwat 28420, Pakistan

**Keywords:** *S-nitrosation*, NMR, thioredoxin, post-translational modification, mechanism of action

## Abstract

Nitric oxide (NO) is a free radical with a signaling capacity. Its cellular functions are achieved mainly through *S-nitrosation* where thioredoxin (hTrx) is pivotal in the S-transnitrosation to specific cellular targets. In this study, we use NMR spectroscopy and mass spectrometry to follow the mechanism of S-(trans)nitrosation of hTrx. We describe a site-specific path for *S-nitrosation* by measuring the reactivity of each of the 5 cysteines of hTrx using cysteine mutants. We showed the interdependence of the three cysteines in the nitrosative site. C73 is the most reactive and is responsible for all S-transnitrosation to other cellular targets. We observed NO internal transfers leading to C62 *S-nitrosation*, which serves as a storage site for NO. C69-SNO only forms under nitrosative stress, leading to hTrx nuclear translocation.

## 1. Introduction

Nitric oxide (NO) is a gaseous free radical with signaling capacity produced in living organisms from bacteria to humans. It is mainly generated from the oxidation of L-arginine catalyzed by NO synthases (NOSs) in most cell and tissue types [1,2]. NO is involved in several fundamental biological processes in the immune, cardiovascular, and nervous systems among other functions [2,3]. Abnormally high levels of NO are toxic and contribute to the development of neurodegenerative diseases, cancer, diabetes, and other pathologies [4,5,6,7,8,9,10,11,12,13].

Cellular functions of NO are in part due to post-translational modification of cysteine residues, known as *S-nitrosation* (also called S-nitrosylation), which results in the formation of a covalent bond between N (NO) and S (Cys). To date, over 3000 proteins are reported as susceptible to *S-nitrosation* [14,15], and this reversible modification can affect protein stability, localization, and activity [4,16]. The main cellular target of *S-nitrosation* is the reduced glutathione (GSH), which is the most abundant thiol in the cell. The S-nitrosated GSH is S-nitrosoglutathione (GSNO), which plays a major role in the S-transnitrosation to other cellular targets. However, GSNO is not efficient in the S-transnitrosation of some targets. One example is caspase-3 (Casp-3), for which the rate constant for S-transnitrosation from GSNO is 120 times lower than from cytoplasmic S-nitrosated human thioredoxin 1 (hTrx-SNO).

Thioredoxin plays an essential complementary role in the S-transnitrosation of specific cellular targets [1]. It is a cytosolic ubiquitous protein that contains five conserved cysteines, including C32 and C35, involved in the redox activity, and C62, C69, and C73 that participate in *S-nitrosation* [17]. A reported example of the thioredoxin *S-nitrosation* target is the catalytic Cys163 of Casp-3. Such modification results in the inhibition of apoptosis [18]. Many hTrx mammalian targets, such as Ras, HIF-1a, X-linked inhibitor of apoptosis, and NF-κB, are involved in the regulation of the cell cycle and other essential physiopathological functions [19,20,21,22,23,24,25,26,27].

Many authors have contributed to the understanding of the NO path within hTrx [17,28,29,30,31,32,33]. The crystal structures of S-nitrosated hTrx showed that C62 is a stable buried target for *S-nitrosation*. The structure of hTrx S-nitrosated at C69 (C69-SNO hTrx) could be determined only at high pH, as C69-SNO is unstable at neutral pHs [1]. Barglow and co-workers have evaluated the influence of the redox state in the site-specific *S-nitrosation* of hTrx by GSNO [29]. They have shown that reduced hTrx was S-nitrosated 2.7 times faster than the oxidized protein. Besides, they have appointed that reduced hTrx was S-nitrosated selectively on C62 and oxidized hTrx (only on C73, the most solvent-exposed cysteine residue) [1,34,35]. It is well accepted that C73 is the most important cysteine for S-transnitrosation [18,30] and that hTrx S-nitrosated at C69 is translocated to the nucleus [36].

In the present work, we used NMR and mass spectrometry to follow the path of S-transnitrosation from GSNO to hTrx. NMR was a powerful tool to detect unstable and intermediate species enabling the description of an in-depth mechanism of S-transnitrosation of hTrx and S-transnitrosation to other targets. Mass spectrometry enabled the unambiguous description of each S-nitrosated form. We measured the reactivity of each of the 5 cysteines of hTrx using mutants with only one cysteine (“C-only mutants”: C32only, C35only, C62only, C69only, and C73only). We also progressively evaluated the *S-nitrosation* of several other mutants containing 2 or 4 cysteines (C62/C69 (C32S/C35S/C73S), C32/C35 (C62S/C69S/C73S), C73S, C35S, and C69S) to understand the wild type hTrx behavior. We found that the redox site is promptly oxidized by the action of GSNO, resulting in no participation of C32 and C35 in the nitrosative activity. We also found an interdependence of the three cysteines in the nitrosative site. We showed that NO follows an entropic path and that C62 is a NO storage site. These findings have an important impact on the understanding of the biological activity of hTrx as an S-transnitrosation agent.

## 2. Materials and Methods

### 2.1. hTrs Constructs and Experimental Design

We used a combination of mutants of human thioredoxin 1 (hTrx) to measure the reactivity of each cysteine of hTrx to direct *S-nitrosation* by GSNO. We designed the C-only mutants that contain only one cysteine and measured the kinetic and reactivity: C32only (C35S/C62S/C69S/C73S), C35only (C32S/C62S/C69S/C73S), C62only (C32S/C35S/C69S/C73S), C69only (C32S/C35S/C62S/C73S), C73only (C32S/C35S/C62S/C69S).

We then analyzed the wild-type hTrx *S-nitrosation* reaction using GSNO as NO donor. To understand the reactivity differences among C-only mutants and Wild type hTrx, we evaluated the *S-nitrosation* kinetics of five other mutants: C32/C35 (C62S/C69S/C73S), C62/C69 (C32S/C35S/C73S), C35S, C73S, and C69S. These were useful to evaluate the possibility of altered reactivity due to the proximity of another cysteine and/or internal transfer of NO. We finally were able to elucidate the mechanism of hTrx *S-nitrosation* using NMR spectroscopy.

### 2.2. Expression and Purification of Human Thioredoxin and Mutants

We used the pET3a vector (GeneScript^®^) containing the sequence of hTrx wild type, C32only, C35only, C62only, C69only, C73only, C62/C69 (C32S/C35S/C73S), C32/C35 (C62S/C69S/C73S), C73S, C35S, and C69S mutants to transform *E. coli* Bl21 (DE3) cells. The isotopic labeling of ^13^C and ^15^N atoms of protein were performed in the M9 medium, where ^13^C_6_-glucose and ^15^NH_4_Cl were used as the sole source of carbon and nitrogen [37]. Protein expression was induced for 18 h using 1 mM of Isopropyl-β-d-thiogalactoside (IPTG) once the cells grew to the mid-log phase (OD_600_ = 0.6). The cells were then pelleted by centrifugation at 4000× *g* for 30 min. The harvested cells were resuspended at 4 °C in phosphate buffer 20 mM, pH 7.0, EDTA 2 mM, DTT 1 mM, NaCl 30 mM, 5 mM, and PMSF 0.1 mM, and then disrupted by sonication (10 cycles of 30 s). The protein supernatant was purified in two steps. First, an ion-exchanged Q-Sepharose column (GE Lifescience) with a salt gradient of 1 M and then by size exclusion chromatography by Superdex 75 resin (GE Lifescience) with phosphate buffer 30 mM, pH 7, NaCl 50 mM, NaN3 5 mM. Figures were prepared with SigmaPlot (SPSS, Inc., Chicago, IL, USA) and BioRender.com (accessed on 23 October 2020). Created with permission from BioRender. Copyright 2022, Vitor S. Almeida.

### 2.3. GSNO Synthesis

The GSNO was prepared by the reaction between GSH and sodium nitrite (NaNO_2_) in an acid medium as mentioned before [38]. After the reaction was accomplished, the pH was set to 6.0 and the solution was stored at −80 °C.

### 2.4. Preparation of C62only(SNO), C69only(SNO), and C73only(SNO) Mutants

We prepared samples of the C62only, C69only, and C73only mutants double-labeled with ^15^N and ^13^C and then subjected them to nitrosative stress by adding to the medium 100 equivalents of freshly prepared GSNO. After two hours of reaction, we removed all GSNO by centrifugation in Merk Millipore Centricon^®^ with a 3 kDa membrane. We performed successive washes with phosphate buffer 30 mM, pH 7, NaCl 50 mM, and NaN_3_ 5 mM until we observed no residual GSNO in the flow-through. Solution absorption spectra of C62only(SNO), C69only(SNO), and C73only(SNO) mutants were acquired after buffer exchange and show a ratio *A*_280_/*A*_335_ (*A*_280_ and *A*_335_ are the Tryptophan and SNO absorption band, respectively) consistent with 1 mol of SNO/mol of protein (data not shown) [1].

### 2.5. NMR Assignment Experiments

Samples of uniformly ^13^C and ^15^N labeled C62only [39], C69only, C73only, mutants, and their respective *S-nitrosation* forms were prepared at a concentration of 400 µM with phosphate buffer 30 mM, H_2_O/D_2_O (90/10%), NaCl 50 mM, 5 mM NaN_3_, and pH 7. For each protein sample, besides the heteronuclear double resonance 2D experiments (^1^H-^15^N HSQC, ^1^H-^13^C HSQC) [40,41] the backbone assignments were achieved by the heteronuclear triple resonance 3D experiments (HNCO, HNCA, HNCACB, HNCOCA, CBCACONH, HBHAcoNH) [42,43,44,45] acquired at 298 K using a Bruker AVANCE-III 800 MHz spectrometer equipped with pulse-field *Z*-axis gradient triple-resonance probe. We collected all 3D data sets using 13% and 15% non-uniform sampling (NUS) mode for the backbone experiments using multidimensional Poisson Gap scheduling [46]. The Reconstruction and processing of the raw data were performed with the hmsIST program that works along with Nmrpipe [46,47,48,49,50,51]. Results were analyzed using the CCPN program (version 2.4.2) [52]. Both are available on the NMRbox platform [53].

We calculated the Chemical shift perturbation (CSP) of ^15^N-^1^H using$\;\Delta \delta ={\left[{\left(\Delta \delta^{HN}\right)}^{2}+{\left(\frac{\Delta \delta^{N}}{10}\right)}^{2}\right]}^{\frac{1}{2}}$ and CSP of ^13^C-^1^H using $\Delta \delta ={\left[{\left(\Delta \delta^{HC}\right)}^{2}+{\left(\frac{\Delta \delta^{C}}{4}\right)}^{2}\right]}^{\frac{1}{2}}$ for the effect of *S-nitrosation* on C62only, C69only, and C73only mutants, where Δ*δ^HN^*, Δ*δ^HC^*, Δ*δ^N^*, and Δ*δ^C^* are the differences between chemical shift, in ppm, observed for the hydrogen bond to nitrogen, hydrogen bond to carbon, nitrogen, and carbon nuclei, respectively. We used the chemical shifts of the backbone atoms (Cα, CO, Cβ, N, Hα, and NH) of the C62only, C69only, and C73only and their respective S-nitrosated forms to calculate order parameters (S^2^) of the secondary structures by the Random Coil Index (RCI) method [54].

### 2.6. NMR Kinetic Experiments

We performed the NMR kinetic experiments with samples of 100 μM of protein double-labeled with ^15^N and ^13^C, 30 mM of phosphate buffer, H_2_O/D_2_O (90/10 %), 50 mM NaCl, 5 mM NaN_3_, and pH 7.0. We determined the protein concentration by UV-Vis spectrophotometry, and the molar extinction coefficient used for all isoforms was 6990 M^−1^ cm^−1^ for absorption with the maximum peak at 280 nm. The reactions were conducted in a regular NMR tube. The tubes were immersed in nitrogen atmosphere. For samples more prompt to oxidation, especially those that contain more than one thiol in the structure, we added a small excess of DTT. The purpose of this addition was to ensure the protein’s completely reduced state at the starting point of kinetics. For the C32/C35 and C73S mutant, the perdeuterated DTT (DL-1,4-dithiothreitol/D10, 98%, CIL) concentration was equal to 200 μM, for the C35S and C69S mutants and the wild hTrx, 300 μM. The small amount of DTT warrants that the reaction starts at the reduced state. Since we use large excess of GSNO in the *S-nitrosation* reaction, the presence of 200 μM of DTT did not interfere with the kinetics. We acquired a ^1^H-^15^N HSQC and ^1^H-^13^C HSQC before adding GSNO at 298 K on a Bruker Avance III 800 MHz spectrometer equipped with a pulsed-field *Z*-axis gradient triple-resonance probe. We used such experiments to characterize the initial condition of the sample (reduced and non-S-nitrosated). Then, we added GSNO freshly prepared to the sample and acquired ^1^H-^13^C HSQC in sequence until the reaction equilibrium. We used varied GSNO concentrations for each mutant to ensure nitrosative stress.

We used the H_β_/C_β_ cross-peak of Cys residues present in the ^1^H-^13^C HSQC before adding GSNO as a reference to 100 μM of protein. To mutants with only one Cys residue, we used the intensity of H_β_/C_β_ cross-peak of Cys of each ^1^H-^13^C HSQC collected after GSNO addition until reaction equilibrium to calculate protein concentration during the reaction. We performed the fitting global and calculated the rate constants of *S-nitrosation* reaction for each mutant with only one Cys residue by KinTek Explorer software [55]. The experimental error analysis was conducted from the global fitting of multiple curves with different concentrations of GSNO. KinTek uses confidence contour analyses, each of the fitting parameters is pushed to higher and lower values while allowing the other variables to be adjusted in deriving the best fit [56]. For all mutants with more than one *S-nitrosation* site as well as hTrx wild type, we used the area of H_β_/C_β_ cross-peak of Cys to calculate protein concentration. We plotted the Cys residues concentration decay to estimate which Cys residue was the most reactive in each mutant evaluated.

We also evaluated the S-transnitrosation reaction between C62only(SNO), C69only(SNO) to GSH. The kinetic data was globally fit using KinTek Explorer. C62/C69(SNO)_2_ mutant also was produced by the same protocol mentioned for C-only(SNO) mutants. Besides GSH, C62C69 (^1^H, ^12^C) and C73only (^1^H, ^12^C) mutants were used as S-transnitrosation targets by C62/C69(SNO)_2_.

### 2.7. Mass Spectrometry Analysis

Samples were diluted to a 1 μM protein concentration in 3% acetonitrile and 0.1% formic acid. 1.0 μL injection volume was loaded on a Waters Nanoacquity system (Waters, Milford, MA, USA). Proteins were desalted online using a trap column (Waters Symmetry C18 180 μm × 20 mm, 5 μm) for 20 min and the LC was performed with an isocratic gradient of 85% acetonitrile containing 0.1% formic acid at a 0.5 μL/min flow in a BEH 130 C18 100 μm × 100 mm, 1.7 μm analytical column (Waters, Milford, MA, USA) for 20 min. The system was set at initial conditions for 20 min to equilibrate the column.

Electrospray mass spectra were recorded using a Q-Tof quadrupole/orthogonal acceleration time-of-flight spectrometer (Waters, Milford, MA, USA) interfaced with the Nanoacquity system (Waters, Milford, MA, USA). The capillary voltage was set at 3500 V, source temperature was 80 °C and cone voltage was 30 V. The instrument control and data acquisition were conducted by a MassLynx data system (Version 4.1, Waters), and experiments were performed by scanning from a mass-to-charge ratio (*m*/*z*) of 300–2000 using a scan time of 1 s, applied during the whole chromatographic process. A 0.1% Phosphoric acid in 50% acetonitrile solution was set at a 0.35 μL/min flow and acquired for 1 s after each 15 s of the main chromatogram to calibrate spectra using Q-Tof’s LockSpray™ (Waters, Milford, MA, USA).

All data were processed manually in MassLynx. To obtain an accurate molecular mass measurement the single resulting chromatographic peak was analyzed and the combined raw mass spectrum was lock mass corrected in MS *m*/*z* scale using phosphoric acid cluster ion 1470.6613 *m*/*z*. The resulting spectra were treated through a charge state deconvolution algorithm-Maximum entropy (MaxEnt 1, Waters, Milford, MA, USA), and average molecular mass was plotted against the resulting spectra’ relative intensity. Peaks were manually assigned to thioredoxin (11,673 Da) or thioredoxin with initial methionine cleaved (11,541 Da), including as modifications nitrosylation (+29.0 Da), oxidation (+16.0 Da), and glutathionylation (+305.1 Da), and allowing for a 1.0 Da tolerance error.

## 3. Results

### 3.1. Reactivity of hTrx Cysteines

We first aimed to determine the *S-nitrosation* reactivity of each cysteine of hTrx. We used mutants containing a single cysteine that we called cysteine-only mutants (C-only). We first evaluated the reaction of the C62only mutant with GSNO (Figure 1a). For all kinetic experiments, we measured the decrease of the intensity of H_β_/C_β_ cross-peak of the reduced cysteine on the ^1^H-^13^C HSQC (Figure 1b) and/or the appearance of H_β_/C_β_ cross-peak from the S-nitrosated cysteine (Figure 1c). Best results were obtained globally fitting the kinetic curves of four different excess concentrations of GSNO. The k_1_ and k_2_ were 5.3 ± 0.2 × 10^−2^ M^−1^s^−1^ and 5.1 ± 2.4 × 10^−2^ M^−1^s^−1^, respectively (Figure 1d). We used mass spectrometry to confirm the products of the C62only *S-nitrosation* reaction (Figure 1e). Despite the high sensitivity, we used the same concentration used for the NMR measurements (Figure 1b,c, 100 µM). We observed the two non-nitrosated C62only species, with (C62SH, 11,673.5 Da) and without the initial methionine (C62SH’, 11,542.5 Da). Over time, we observed the formation of the S-nitrosated species (C62SNO and C62SNO’) [40].

We also measured the C69only (Appendix A), C73only (Appendix A), C35only (Appendix A), and C32only (Appendix A) *S-nitrosation* kinetics. The forward rate constant for C69only was 1.9 ± 0.1 × 10^−1^ M^−1^s^−1^, 3.6 times bigger than for C62only. The reverse rate constant was 3.8 ± 0.6 × 10^−1^ M^−1^s^−1^. Different from C62only, we also observed S-glutathionylation for C69only (Appendix A). Remarkably, the reaction of the C73only with GSNO led to *S-nitrosation* and two thiolation reactions: dimerization and S-glutathionylation. We confirmed the dimeric product formation by SDS-PAGE with different concentrations of GSNO (Appendix A). The relative monomer/dimer ratio varied with the excess of GSNO. For conditions of nitrosative stress (10× and 20× of GSNO), the mutant was mostly monomeric, whereas it was not possible to determine by this experiment how much of it was S-nitrosated and how much was S-glutathionylated. It is important to note that the ratio between monomer and dimer remained practically unchanged over time when comparing two experiments carried out with 20 times excess of GSNO with reaction times of 1 h and 20 h.

To characterize the post-translational changes that occurred in the C73only mutant upon reaction with GSNO, we carried out a mass spectrometry analysis (Appendix A). Several species were detected: (i) unmodified C73SH with and without the initial methionine (11,673.3 Da and 11,542.4 Da, respectively); (ii) S-nitrosated C73only (+29 Da); and (iii) S-glutathionylated C73only (+305 Da), both observed after 1200 s of reaction. At 7200 s of reaction, we observed an increase in the relative intensity of the peaks corresponding to the S-glutathionylated state. We obtained 2.5 ± 0.5 M^−1^s^−1^ as the forward rate constant (k_1_) and 0.9 ± 0.5 M^−1^s^−1^ as the reverse rate constant (k_2_) for the *S-nitrosation* reaction of C73only. We also determined the forward rate constant for the dimerization reaction (k_3_) as being 2.8 ± 1.3 M^−1^s^−1^.

For the mutants with only one cysteine, one *S-nitrosation* site, C32only (Appendix A), and C73only were the most reactive followed by the C69only. C35only (Appendix A) presented a reactivity very similar to that of the less reactive mutant, C62only. Figure 2 shows the comparison among the forward rate constants for the *S-nitrosation* reactions of each mutant by GSNO. 

### 3.2. S-nitrosation Kinetics of Wild Type hTrx

The next step was the measurement of the *S-nitrosation* kinetics of the wild type hTrx to investigate the cross-reactivity of the three cysteines in the *S-nitrosation* site and 2 cysteines in the redox site. We followed independently the C_β_H_β_ resonances of each cysteine (Figure 3a). Both C32 and C35 cross-peaks in their reduced form vanished from the spectra immediately after the reaction started. The immediate oxidation of the redox site in the presence of GSNO was also demonstrated in the experiments with the mutant containing only C32 and C35 (C62S/C69S/C73S, C32/C35) (Appendix A). This oxidation could be the result of the *S-nitrosation* of C32 (the most reactive, Figure 2), followed by the nucleophilic attack of C35 thiolate to the sulfur of C32-SNO. However, this result must be analyzed with caution, since contamination of oxidized glutathione (GSSG) could also lead to this immediate oxidation.

A remarkable finding was that rate of disappearance of reduced resonance of C69 and C73 was much slower than that observed for C69only and C73only. For C73, compare the curves of 10× GSNO of the wild-type (Figure 3) and C73only (Appendix A). For C69, compare the curves of 10×, 20× and 30× of GSNO of the wild-type (Figure 3) and C69only (Appendix A). On the other hand, the cross-peaks of the reduced C62 decayed much faster than the C62only mutant (Figure 3B). These data suggest that there is an interdependence among the nitrosative cysteines in hTrx. To better understand this phenomenon, we constructed mutants containing two and three cysteines of the nitrosative site: C62/C69 (C32S/C35S/C73S), C69S, C32/C35 (C62S/C69S/C73S), C73S, C35S, and C35S/C73S. We will refer to this inversion of decay rates observed for C69only and C62only mutants (C69 decays faster than C62) compared with the wild type (C69 decays slower than C62) as a C69-C62 inversion. This inversion is essential to our hypothesis.

We detected the products of the *S-nitrosation* reaction of wild-type hTrx by mass spectrometry (Figure 3C). The data confirmed the presence of mono (12,696.0 Da), double (12,725.0 Da), and triple (12,754.0 Da) S-nitrosated wild type hTrx. We also observed the presence of the protein with one glutathionylation (hTrxG) and two S-glutathionylation (hTrxGG). We observed mixed states such as mono-S-nitrosated state and S-glutathionylated (hTrxNOG) and bi-S-nitrosated and mono S-glutathionylated (hTrxNONOG).

### 3.3. Mutants with Multiple S-nitrosation Sites to Define the Path of NO

To investigate the interdependence among different *S-nitrosation* sites, we evaluated the *S-nitrosation* kinetic of the other four hTrx mutants with multiple *S-nitrosation* sites. For the mutant C62/C69, we observed the same C69-C62 inversion as for the wild-type hTrx. C62 behaves similarly, while C69 decayed slower than that observed for C69only. These results suggest either an internal transfer of NO from C69 to C62 or negative cooperativity between C62 and C69 (Appendix A).

The mutant C35S mimics the *S-nitrosation* behavior of the wild-type reduced state and by studying the *S-nitrosation* reaction using C35S we could test the redox-control hypothesis [30] over the *S-nitrosation* reaction. For the wild type, we observed immediate oxidation at the redox site, forming a disulfide C32/C35. However, the contamination of GSSG in our preparation of GSNO could contribute to the oxidation of the redox site, preventing any conclusion regarding the redox control. The contamination of GSSG does affect redox site of the mutant C35S and the reduced form of C35S hTrx behaved similarly to the wild type in terms of the response to *S-nitrosation* (Appendix A). This data suggests that the oxidation of the redox site does not interfere significantly in the nitrosative site, meaning that a redox control in the *S-nitrosation* reaction is less likely [30]. Note that for the 10× excess of GSNO at ~10.000 s, C62 cross-peaks remained ~30% of its area and C73 remained ~5%, for both C35S and wild type hTrx. C69 cross peak remained ~35% for C35S and ~50% for the wild type hTrx. Altogether, we showed that the *S-nitrosation* reaction is similar, independently of the oxidation state of the redox site.

To evaluate the relative contribution of C73 to the nitrosative site, we studied the kinetics of the mutant C73S. Remarkably C73S, C62 and C69 decayed slower when compared to the wild type (Appendix A). Note that the C69 cross peak remained ~100% of its intensity after ~10.000 s (10× excess of GSNO) while the wild type decayed 50%, suggesting the importance of C73 as the most reactive cysteine (Figure 2). For C62 resonance, the decay was ~65% for C73S and ~30% for the wild type. We also observed the C69-C62 inversion for C73S, and, as with the wild type, the redox site also oxidized immediately after the addition of GSNO.

Another important point to take into consideration was the possibility of an intermolecular NO transfer. To address this, we compared the *S-nitrosation* of the mutant C35S ({^15^N, ^13^C}-labeled C35S, 100 µM) in the presence and absence of the mutant C73only (unlabeled, 100 µM) (Appendix A). We did not observe a statistically significant difference between the two conditions, suggesting that since doubling the concentration of C73 did not change the response of the system an intermolecular external NO transfer would not play any role.

C62 is the less reactive cysteine in the nitrosative site (Figure 2, C62only) and is involved exclusively in *S-nitrosation*. For the wild type hTrx, the C62/C69, and C35S mutants, C62 was not involved in thiolation reactions, either glutathionylation or disulfide formation. For this reason, we will argue that C62 is the ultimate destination of the NO, working as a NO storage site. The other consideration to classify C62 as the storage site is the C69-C62 inversion. C62 is also the least exposed [1]. The importance of C73 to the effectiveness of the *S-nitrosation* was also established. With these data in mind, two questions remained to be answered: what is the importance of C73 in the NO transfer to the storage site? Does it go directly to C62 or via C69 in the path? To answer these questions, we measured the *S-nitrosation* kinetics of the mutant C69S.

The mutant C69S was essential to show the importance of C73 to *S-nitrosation*. In this mutant, the decay rates are similar to the wild type for either C62 or C73. Note that for the 10× excess of GSNO at ~10.000 s, cross-peaks correspondent to C62 remained ~35% of its area for C69S and ~30% for the wild type, C73 remained ~15% for C69S and ~5% for wild type (Figure 3b and Appendix A).

For the mutant C69S, we did not observe a C73-C62 inversion. However, C73 decay is much slower for the C69S mutant than that observed for C73only. Different from the mutant C62/C69, where C62 decay was similar to the C62only mutant, for C69S, the C62 peak area decays faster than for C62only, meaning that the effect of the presence of C73 increased the *S-nitrosation* of C62, being more efficient than the effect resulting from the presence of C69. Note that in the presence of C69 (C62/C69 mutant), while the rate of C69 *S-nitrosation* became lower than that observed for C69only mutant, the rate of C62 *S-nitrosation* was the same as observed for C62only mutant. In Figure 4, we compared the peak area decay of the reduced C62 in the wild type and 5 hTrx mutants. When comparing 10x excess of GSNO, we show that the presence of C73 in wild type hTrx (Figure 4a), C35S (Figure 4b), and C69S (Figure 4c) mutants displayed a faster decay of the reduced C62 when compared to C62only (Figure 4d), C62C69 (Figure 4e) and C73S (Figure 4f), in which C73 are absent. Therefore, C73 is responsible for potentializing *S-nitrosation* of C62, especially at a lower excess ratio of GSNO.

### 3.4. Fine Details in the Changes in Protein Structure by Nitrosation

*S-nitrosation* has the chance to promote changes in the structure and dynamics of a protein. We observed a fine-tuned regulation of the protein dynamics upon *S-nitrosation*. For this, we calculated the random coil index, interpreted as the chemical shift-based order parameter, S^2^ [57] from the resonance assignment of the backbone of C62only and S-nitrosated C62only (C62onlySNO) (Figure 5a), C69only and S-nitrosated C69only (C69onlySNO) (Figure 5b) and C73only and S-nitrosated C73only (C73onlySNO) (Figure 5c).

#### 3.4.1. General Flexibility Changes after Nitrosation as Observed by S^2^ Measurements

We observed that the S nitrosation led to changes in the flexibility of the β3/α3 loop and N-terminal region of α3 that contains C62, and α3/β4 loop containing C69 and C73. The random coil index takes into consideration the chemical shifts of the backbone (Cα, CO, Cβ, N, Hα, and NH). The lower the S^2^ the greater the flexibility. From the *S-nitrosation* of residue C62, the region comprising residue S73 became more flexible, given a decrease in S^2^. The α3 helix, on the other hand, showed greater rigidity when residue C62 was S-nitrosated. C69 *S-nitrosation* resulted in a small increase in flexibility of α3 and α3/β4 loop. C73 *S-nitrosation* resulted in a remarkable rigidity increase at α3/β4 loop, which characterizes an entropically unfavorable process.

The dynamics of all C-only mutants were similar, except for C73only, with was more flexible at the α3/β4 loop. The loops that render mobility to the nitrosative site are β3/α3 and α3/β4. C62 is at β3/α3 at the N-cap of α3, C69 is at the C-cap of α3 and C73 is at the α3/β4 loop. Changes in flexibility in these regions are relevant to the nitrosative site. C73 of the non-nitrosated C73only is in the most flexible loop of the protein (Figure 5c,d). Upon *S-nitrosation*, C73 became more rigid with S^2^ going from 0.69 to 0.83. Thus, despite *S-nitrosation* at C73 being kinetically favorable (and the most reactive cysteine, Figure 2), the increase in rigidity makes it unfavorable in terms of the conformational entropy (α3/β4 loop became more rigid). The *S-nitrosation* of C62 led to a subtle increase in the order parameter of β3/α3 loop. The *S-nitrosation* of C69only resulted in subtle changes in S^2^. These data suggest that conformational entropy may play a role in the *S-nitrosation*. Denitrosation at C73 and C62, especially at C73, seems to be conformationally entropically favorable and may favor internal transfers.

#### 3.4.2. Changes around Nitrosated Sites as Analyzed by CSP

We also analyzed the chemical shift perturbation of ^1^H-^15^N (^1^H-^15^N CSP, Appendix A) and cysteine ^1^H_β_-^13^C_β_ (Figure 6) to understand the structural and dynamical features at the nitrosative site. The *S-nitrosation* of C62only led to changes in ^1^H-^15^N CSP in all cysteines at the nitrosative site, including β3/α3 loop, α3, and the α3/β4 loop (Appendix A), showing cooperativity between C62 and most of the residues of the nitrosative site. *S-nitrosation* of C69only elicited ^15^N-^1^H CSP confined to the C-terminal region of α3 and the α3/β4 loop (Appendix A). For C73only ^15^N-^1^H CSP was local, only at adjacent residues (Appendix A). These results can be explained by the fact the *S-nitrosation* of the buried residue C62 led to more significant structural changes compared to C69 and C73, in this order. The important conclusion is that the *S-nitrosation* of C62 led to structural changes in the whole nitrosative site but *S-nitrosation* of either C69 or C73 did not affect significantly C62.

The analysis of ^1^H_β_-^13^C_β_ CSP was elucidative since it is near the chemical environment of SNO (Figure 6). The large ^1^H_β_-^13^C_β_ CSP observed for C62 resulted from the oxidation of the redox site and not from the *S-nitrosation* of either C73 or C69. This agrees with the analysis of the ^1^H-^15^N CSP. The *S-nitrosation* of wild type hTrx, C69S, and C73S, all of them containing the intact redox site, led to significant chemical shift changes at C62, whereas the *S-nitrosation* of C35S and C62C69, incapable of forming a disulfide bond at the redox site, did not show significant chemical shift changes at C62.

Large ^1^H_β_-^13^C_β_ CSP at C69 and C73 correlates with the *S-nitrosation* of C62 and not to oxidation of the redox site. The *S-nitrosation* of C62 led to a significant ^15^N-^1^H CSP (Appendix A) at the C-terminal portion of α3 and α3/β4 loop, regions that contain C69 and C73. The analysis of ^1^H_β_-^13^C_β_ CSP reinforced this observation. The *S-nitrosation* of wt hTrx, C35S, C73S, and C62C69 (Figure 6), all led to chemical shift changes in C69. All of them have C62 present, while the redox site is incapable to form the disulfide bond.

As mentioned earlier, the presence of C73 has a remarkable effect on the kinetics of C62, making the *S-nitrosation* of C62 much faster. We observed ^15^N-^1^H CSP for the loop containing the C73 when C62only was S-nitrosated (Appendix A). The analysis of ^1^H_β_-^13^C_β_ CSP showed that C73 shifted for wt hTrx, C69S, and C35S (Figure 6). The absence of C69 in the mutant C69S showed that the observed chemical shift change in C73 is not due to the presence of C69. Similarly, the chemical shift changes in C73 were not due to disulfide bond formation in the redox site since a similar shift was observed for the C35S mutant. Most probably, ^1^H_β_-^13^C_β_ CSP of C73 is due to the *S-nitrosation* of C62, as observed for the mutant C62only (Appendix A).

### 3.5. S-transnitrosation of hTrx-SNO to GSH and hTrx Mutants

So far, we have described the pathway for the S-transnitrosation from GSNO to hTrx. Now, we wanted to probe the S-transnitrosation of hTrx-SNO to other targets. This is a complex question since it may also depend on the target. However, GSH is a good target to mimic a thermodynamically favorable S-transnitrosation to an exposed cysteine, as GSNO is one of the most stable nitrosothiols. We probed the S-transnitrosation reaction by preparing C62only-SNO, C69only-SNO, and C62/C69-(SNO)_2_ (Appendix A). For C62/C69-(SNO)_2_, C62 is fully S-nitrosated, while C69 is only ~75%. We globally fit the *S-nitrosation* kinetic from C62only-SNO to GSH (Appendix A). For the global fitting, we had to include the formation of C62only-glutathionylated (C62only-G) in very small amounts. S-nitrosated species were confirmed by mass spectrometry. The reaction rate for S-transnitrosation for C62only-SNO was 5.9 ± 0.6 × 10^−2^ M^−1^s^−1^ and 4.9 ± 1.2 × 10^−2^ M^−1^s^−1^ for glutathionylation, which occurred in very small amounts (Appendix A). Note that the reaction is slow to be biologically relevant in the cellular environment, which is consistent with the role of C62 as a storage site. We showed similar behavior for C62 in the S-transnitrosation reaction from C62/C69-(SNO)_2_ (Appendix A).

We also probed the S-transnitrosation from C69-SNO to GSH for the C69only-SNO and C62/C69-(SNO)_2_. C69 was more efficient than C62 for S-transnitrosation (14 times faster), but, for both mutants, the reaction was slow, making C69 unlikely to be an efficient site for S-transnitrosation in hTrx (Appendix A). We globally fit the *S-nitrosation* kinetic from C69only-SNO to GSH (Appendix A). Here also we had to consider the formation of C69only-glutathionylated (C69only-G) in very small amounts. The reaction rate for the S-transnitrosation for C69only-SNO was 8.4 ± 1.2 × 10^−1^ M^−1^s^−1^ and 2.3 ± 1.0 × 10^−1^ M^−1^s^−1^ for glutathionylation, which occurred in very small amounts (Appendix A). Although it happened on a very small scale, the formation of C69only-G was ~5 times faster than for the C62only-G, being C69 more efficient than S-glutathionylation. The mass spectrometry experiment confirmed the S-glutathionylated species.

This observation is consistent with C73 being the most efficient site for S-transnitrosation. Barglow et al. (2011) [29] proposed the external intermolecular S-transnitrosation from C62-SNO to C73, making C73-SNO prone to S-nitrosate to other targets. We probed the S-transnitrosation from C62/C69-(SNO)_2_ to unlabeled C62/C69 (Appendix A) and also to unlabeled C73only (Appendix A), both intermolecular NO transfer. In both situations, we did not observe the S-transnitrosation reaction. This data follows the previous conclusion that the internal NO transfer from C62-SNO (storage site) is the main pathway to form the C73-SNO, prone to S-transnitrosate to multiple targets.

## 4. Discussion

In the present work, we used several mutants of the cytoplasmic human thioredoxin 1 (hTrx) to decipher the path of SNO and the relative contribution of each of the five cysteines. We first described the reactivity and chemoselectivity of each cysteine and established the basis for the discussion. Most importantly we showed that C62 is a slow, end-of-process NO storage site in hTrx. C73 is the most reactive and probably the first S-nitrosated state, which can transfer NO intramolecularly to C62. We also showed that intermolecular transfers of NO are negligible. Based on NMR chemical shift differences, C62 nitrosation impacts hTrx structure more than C69 or C73 nitrosation (without changing the global fold).

One important discussion that comes along with the proposition of the internal NO transfer is its plausibility since, for most of the hTrx structures, C62 and C73 are more than 10 Å apart. A direct nucleophilic attack from C62 in a thiolate form to the electron-rich nitrogen of the C73-SNO is necessary for this NO internal transfer. The possibility of an external intermolecular transfer between two hTrx molecules was excluded (Appendix A). The answer is in the plasticity of the nitrosative site. The low order parameters at the α3/β3 loop, the N-terminal of α3, and mainly at the α3/β4 loop showed that there is flexibility at the nitrosative site (Figure 5). Hwang et al. (2015) showed the crystal structure of a fully oxidized hTrx (5DQY), with the formation of a disulfide bond between C62 and C69, despite being more than 10 Å apart and separated by α3 in all other available structures [58]. The formation of this disulfide bond implies α3 unfolding. For the disulfide bond formation, it is presumed a helix/coil equilibrium for α3 to enable the direct nucleophilic attack between these two residues. All of this together makes the internal NO transfer structurally and dynamically plausible.

Another point must be taken into consideration, which is the chemical mechanism able to explain the chemo and regioselectivity promoted by protein for the S-nitrosothiol reactions. Thalipov and co-workers [59] computational calculations successfully described the reaction mechanism and the resonance structures of a S-nitrosothiol (S, D or I, Figure 7). The mechanism explains how the local environment (regioselectivity) of each cysteine in the protein can control if the site is more susceptible for *S-nitrosation* or thiolation (chemoselectivity). The weak covalent bond energy of only ~30 kcal/mol and the long S-N bond (~1.8 Å) was explained by the contribution of an ionic resonance structure ***I***. The double bond character of the S-N bond arises from the zwitterion ***D***. The thiolation/*S-nitrosation* results from the relative contribution of ***S*** and ***D*** resonance forms [59,60]. Our data indicate a protein control that differentiates the three cysteines of the nitrosative center of hTrx. While C62 could only S-nitrosate, C69 and C73 could S-nitrosate (resulted from a nucleophilic attack to the ***S*** form) and thiolate (resulted from a nucleophilic attack to the ***D*** form). C73 thiolation led to S-glutathionylation and dimerization, while C69 only to S-glutathionylation. Moran and co-workers [60] studied the hydrolysis of S-nitrosothiols in which the nucleophile is water. Their calculation shows the formation of an ionic intermediate (probably forming an intimate ion pair between RS^−^ and NO^+^) that favors the nucleophilic substitution, probably via S_N_1-like. We propose here that the presence of an ionic resonance structure ***I*** favors a longer distance nucleophilic attack where NO^+^ would have relative freedom at the nitrosative site and the protein structural and dynamical demand for the internal transfer would not be high.

In all experiments containing the intact redox catalytic center, we observed rapid disulfide bond formation between C32 and C35. This observation could be a result of the rapid *S-nitrosation* of C32 followed by thiolation and conversion to a disulfide bond between C32 and C35. Could also be an artifact due to the contamination of GSSG to the GSNO preparation. Nevertheless, our experiments showed unambiguously that the reduced and oxidized forms of the two catalytic cysteines (C32 and C35) did not impact the *S-nitrosation* of the three other cysteines and/or the NO transfer between them. The similar *S-nitrosation* activity of wild type hTrx and C35S (mimics the reduced state) impacts the widely accepted interpretation that the oxidized and reduced hTrx have different S-transnitrosation activities [30], we suggested that the oxidized and reduced states have similar *S-nitrosation* activity, and the redox control may not be relevant. Moreover, there is no biological role described so far for the reduced S-nitrosated hTrx [61].

Based on the kinetic and structural data, we propose a biological mechanism for hTrx *S-nitrosation* (summarized in Figure 8), depicting the role of the many post-translational modified species of hTrx detected here. We took into consideration that our studies used a concentration of hTrx and GSNO much higher than the cellular content. Of note, the main findings in this work are independent of the concentration, such as the observed reactivities of the individual cysteines and the presence of internal NO transfer. The relative amount of the S-nitrosated, glutathionylated, and dimeric species is largely dependent on concentration and this dependence will be discussed in the next paragraphs. In the proposed mechanism we combined the data of the present work and the vast biological data available in the literature (Figure 8). The measurement of the pKa values of the different thiols could further clarify the reactivity of the cysteines to GSNO.

The S-nitrososative cycle starts with the oxidation of the redox site, in which the redox state of hTrx depends on the redox potential of the cytoplasm controlled by the expression level of thioredoxin and the NADPH available. We assumed, based on the results discussed earlier, that the redox state of hTrx will not change the following outcome of events. C73 was the most reactive cysteine and supposedly is the first S-nitrosated state to be formed. C73-SNO is the state responsible for S-transnitrosation to other targets, such as caspase 3 [17]. An alternative path for NO is the internal NO transfer to C62 producing a low reactive species (Figure 8, in yellow), referred to here as the NO storage site of hTrx. C62 *S-nitrosation* is kinetically unfavorable but thermodynamically favorable. We propose that NO follows an entropic favorable path and entropy compensation may play a major role in the internal transfer.

An important discussion that comes from our data is the biological relevance of the reduced state of C62-SNO hTrx. In the presence of a reduced environment in the cell (high NADPH) C62-SNO hTrx can be reduced by the thioredoxin reductase (hTrxR)/hTrx system and be present. Further studies are necessary to understand the role of reduced C62-SNO hTrx, but in our point of view, it is likely that C69 and C73 are promptly denitrosated by the hTrxR/hTrx system, while C62 is more resistant to S-denitrosation, functioning as a stock and acting as a buffer of SNO in the cell in both reduced and oxidized environment. The thermodynamically stable C62-SNO storage site could act as a buffer of SNO under nitrosative stress conditions [66,67]. The biological role of hTrx in S-denitrosation is of major relevance and is one of the main functions of hTrx [2]. *S-nitrosation* in the cell is an important physiological signaling pathway, such as phosphorylation, and nitrosative stress is behind several degenerative diseases [2,16]. hTrx overexpression in Hela cells has been demonstrated to be protective against nitrosative stress. Smith and Marletta [61] pointed out the lack of evidence that the reduced form of C62-SNO hTrx can S-transnitrosate a cellular target. By contrast, oxidized C73-SNO hTrx can S-transnitrosate to multiple targets [61]. Maybe, the reduced form of C62-SNO hTrx is not relevant for S-transnitrosation but mainly serves as NO storage.

The present data also suggest that C62 is not subject to direct *S-nitrosation* from GSNO. C62 is buried and direct *S-nitrosation* was slow compared to the *S-nitrosation* of either C73 or C69 followed by the internal NO transfer to C62. C62-SNO and C69-SNO were also unable to efficiently S-transnitrosate to GSH directly. In our opinion, reverse NO transfer from C62 to C73 is necessary for the effectiveness of C73-SNO to S-transnitrosate other cellular targets (Figure 8). C62, as the storage site, buffers the influence of the cellular environment in the S-transnitrosation activity of hTrx.

C73 is the most solvent-exposed cysteine of hTrx and is well known for being responsible for all described S-transnitrosation cellular events involving hTrx [2,68]. Many mammalian targets are involved in cell cycle regulation or other essential physiopathological functions. Among these proteins are caspases 3, Ras, HIF-1a, X-linked inhibitor of apoptosis, and NF-κB [19,20,21,22,23,24,25,26,27]. C73 is also the site for in vitro thiolation of hTrx, leading to dimerization and S-glutathionylation (Figure 7). So far, the physiopathological role of thiolation in the cell is not fully understood. A small amount of GSNO formed almost equivalent quantities of S-nitrosated and dimeric hTrx. At these concentrations of GSNO, we also observed the formation of the glutathionylated and S-nitrosated species. The glutathionylated-hTrx (hTrx-G) is formed by the attack of GSH on the SNO-hTrx. The increase in the excess of GSNO favored the formation of the monomeric hTrx, probably due to the higher propensities for the formation of the S-nitrosated-hTrx, preventing the formation of the dimeric form. Our data suggest that hTrx thiolation should be significant only at basal levels of GSNO and less important under nitrosative stress. Wu et al. (2011) [69] reported that the cellular level of SNO-hTrx is tightly regulated by the presence of reduced hTrx and GSH, both present under physiological conditions. Among the thiolated species, the in vitro experiments suggested a more major proportion of dimeric hTrx than hTrx-G. This may not be true in a cellular environment due to the crowding effect. Among the hTrx functions in the cell, the fine-tuning between *S-nitrosation*, NO storage, S-denitrosation, and thiolation is fundamental for the cell redox and nitrosative homeostasis. The thiolated species regulate both the amount of the NO storage state (C62-SNO) and the reactive form (C73-SNO) of hTrx.

We also showed that the *S-nitrosation* of C62 has a negative cooperative effect on the *S-nitrosation* of C69. Therefore, *S-nitrosation* of C69 only occurred at high levels of GSNO. Nitrosative and oxidative stress are directly related to hTrx nuclear translocation (Figure 8), activating the expression of transcription factors related to cell survival and proliferation [68]. At the same time, nitrosative stress leads to S-transnitrosation of the active site of caspase-3, inhibiting apoptosis (Figure 8). While *S-nitrosation* of caspase-3 and other cellular targets occurs through C73 [2,12,18], the nuclear translocation occurs through the *S-nitrosation* of C69 in a karyopherin-dependent manner [36]. In Figure 8, the arrows in red refer to reactions that occur only at nitrosative stress. The negative cooperative effect of C62 is important to downregulate the *S-nitrosation* of C69 and warrant that it will occur only under nitrosative stress conditions. All data indicate that C69-SNO does not S-transnitrosate to other targets. This role is reserved for C73.

The functions of hTrx as a NO storage site make it important to regulate, along with the thiolated states, the reactivity of hTrx. At physiological conditions with low levels of GSNO, the NO will be preferentially located at C62, being C62-SNO thermodynamically stable with low reactivity (Figure 8). Under nitrosative stress (Figure 8, arrows in red), at high levels of GSNO, the levels of C73-SNO and C69-SNO are high, decreasing the level of thiolated species (Appendix A).

In the experimental design of the present work, we constructed mutants that enabled the observation of intermediate states of SNO-hTrx. NMR enables the observation of site-specific reactions and their kinetic and structural consequences. This was powerful in determining the NO path in hTrx. The C-only mutants gave information on the reactivity of each cysteine individually. We gradually increased the complexity of the mutants, which enabled the observation of the interrelation between the different cysteines and the internal NO transfer. There are two major limitations to our strategy, namely the use of high concentrations demanded by NMR and the timescale of the experiments. To overcome the concentration problem, we used mass spectrometry to show unambiguously the species formed under identical experimental conditions of the NMR kinetic experiments. The timescale limited our capacity to describe all of the rate constants and globally fit the described model.

In summary, we described in detail the path of NO transfer to and from hTrx (Figure 7). The interpretation of the experimental data combined with the vast biological data available in the literature enabled us to propose a mechanism for the nitrosative cellular cycle. The physiological path begins in the nitrosative site of hTrx with the prompt formation of C73-SNO followed by internal transfer to C62, which serves as a NO storage site. The redox site is promptly oxidized and does not regulate the nitrosative activity. To modify hTrx cellular targets the NO is internally transferred back from storage (C62-SNO) to C73 and, finally, to the target protein. C69 is not S-nitrosated under physiological conditions since C62-SNO displays negative cooperativity towards C69. Remarkably, C69 is modified only under nitrosative stress and serves as a signal to hTrx translocation to the nucleus, thus, regulating the cell responses and inducing a proliferative state.

## Figures and Tables

**Figure 1 antioxidants-11-01236-f001:**
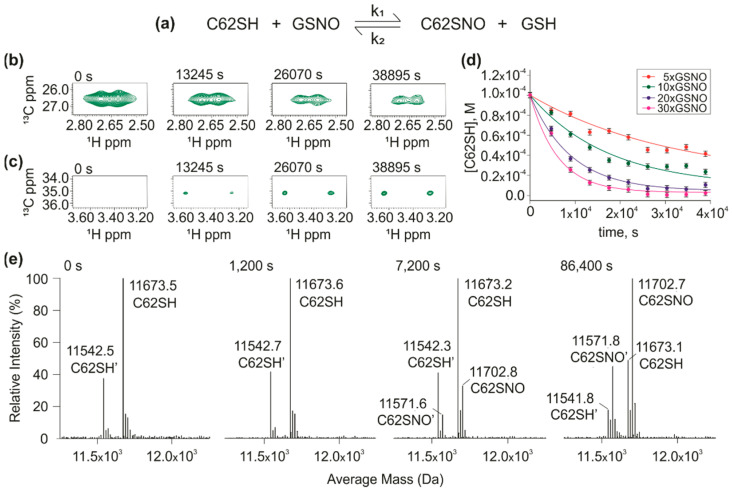
*S-nitrosation* kinetic of C62only (C62SH): (**a**) The reaction between C62SH mutant and GSNO; (**b**) ^1^H-^13^C HSQC of C62SH (100 µM) and 10× of GSNO reaction at different reaction times, as an example. They show H_β_/C_β_ cross-peak of C62SH intensity decreasing over time; (**c**) At the same time, the ^1^H-^13^C HSQC shows the H_β_/C_β_ cross-peak of C62SNO intensity increasing over time; (**d**) Global fitting of kinetic data calculated based on H_β_/C_β_ cross-peak of C62SH. The curves correspond to different reaction conditions: 5 (500 µM-red), 10 (1000 µM-green), 20 (2000 µM-blue) and 30 times of GSNO (3000 µM-magenta). Rate forward constant (k_1_) calculated for the C62SH *S-nitrosation* reaction was 5.3 ± 0.2 × 10^−2^ M^−1^s^−1^, reverse one (k_2_) was 5.1 ± 2.4 × 10^−2^ M^−1^s^−1^; (**e**) Mass Spectra of reaction between C62SH (100 µM) and 10x of GSNO acquired in different reaction times (0 s, 1200 s, 7200 s, and 86,400 s). C62SH without first Met is shown as C62SH’. C62SNO and C62SNO’ are S-nitrosated species. The experimental error for each experimental point in D was estimated from the standard deviation of the spectral noise for one experiment.

**Figure 2 antioxidants-11-01236-f002:**
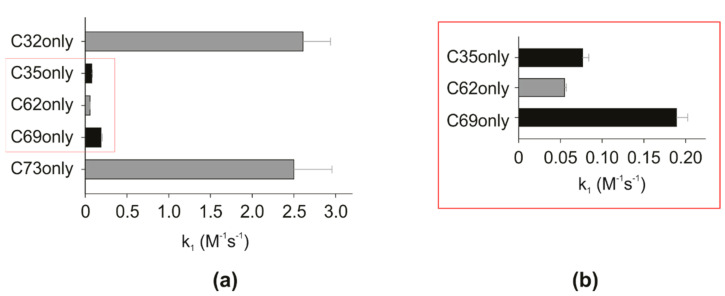
*S-nitrosation* reaction rate constants for each mutant: (**a**) C32only and C73only mutants stand out as the most reactive in the presence of GSNO; (**b**) the 10× zoom of the general graph, which makes it possible to observe the less reactive mutants C35only, C62only, and C69only. Mutants order according to decreasing reactivity is C32only, C73only, C69only, C35only, and C62only. The experimental error analysis was conducted from the global fitting of multiple curves (*n* = 1) with different concentrations of GSNO. The error bar refers to the standard deviation (SD).

**Figure 3 antioxidants-11-01236-f003:**
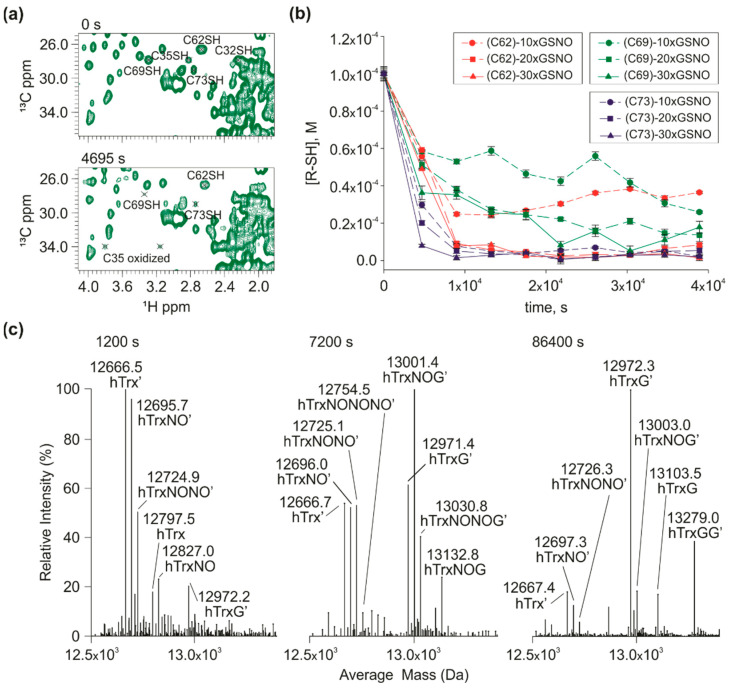
*S-nitrosation* kinetics of wild type hTrx: (**a**) ^1^H-^13^C HSQCs acquired before (0 s) and after (4695 s) the addition of 10× excess of GSNO. In the upper zoom, it is possible to observe the cross-peak H_β_/C_β_ of the non-nitrosated residues C32, C35, C62, C69, and C73. After 4695 s, only the signals of residues C62, C69, and C73 can be observed. The signals of C32 and C35 do not appear in the spectrum in their reduced forms, after the addition of GSNO. It is possible to observe only the oxidized C35; (**b**) The red, green, and blue curves correspond to the *S-nitrosation* kinetics of cysteines C62, C69, and C73, respectively, with an excess of 10×, 20×, and 30× GSNO. The concentration of each non-nitrous cysteine residue was monitored over time. C73 is the most reactive residue, as it showed the greatest decay; (**c**) Mass spectra were acquired after the addition of an excess of 10x of GSNO at three different reaction times: 1200 s, 7200 s, and 86,400 s. At 1200 s, we observed the formation of the doubly S-nitrosated species hTrx’.NO.NO and mono glutathionylated hTrx’, in addition to the presence of the species mono S-nitrosated (hTrx.NO and hTrx’.NO.G. After 7200 s of reaction it was possible to observe the same protein accumulating three modifications, two *S-nitrosations* plus one S-glutathionylation (hTrx’.NO.NO.G) and three *S-nitrosations* at the same time (hTrx’.NO.NO.NO). After 86,400 s of reaction the major peaks in the spectra are equivalent to a protein mono and doubly S-glutathionylated, hTrx’.G and hTrx’.G.G, respectively. The experimental error for each experimental point in B was estimated from the standard deviation of the spectral noise for one experiment.

**Figure 4 antioxidants-11-01236-f004:**
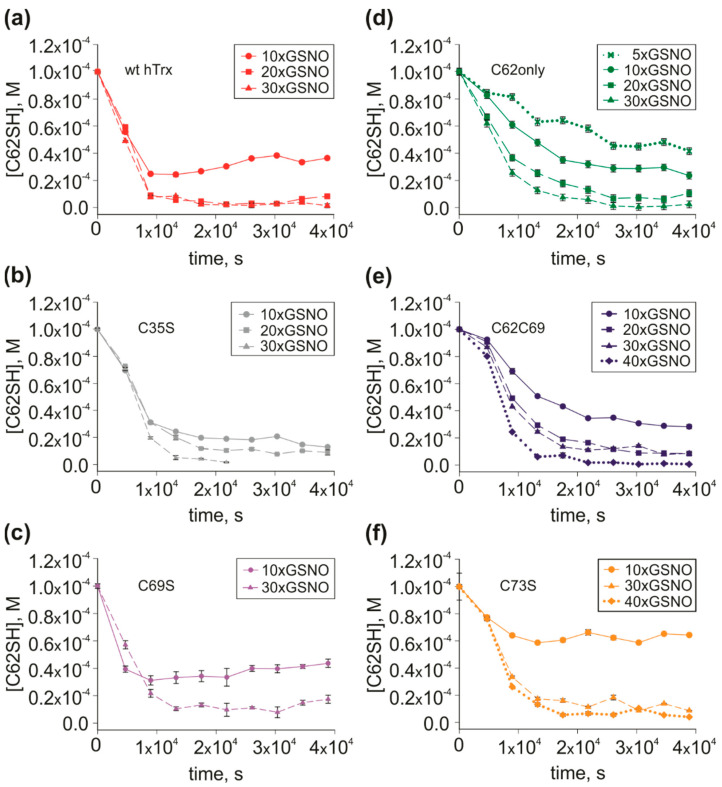
C62SH concentration decay. *S-nitrosation* kinetics of hTrx mutants (100 µM) with increasing GSNO concentration. C62SH concentration was calculated based on the C62SH signal area. On the left, C62SH concentration decay of wild type hTrx (**a**), C35S (**b**), and C69S (**c**) mutants. The oxidation of C62SH in the mutants is similar to that in wild type hTrx. On the right, C62SH concentration decay of C62only (**d**), C62C69 (**e**), and C73S (**f**) mutants showing a slower decay if compared with C62SH in wild type hTrx. These observations highlight the importance of C73 to C62 *S-nitrosation* and demonstrate that oxidation of the redox site and C69 do not affect the C62 *S-nitrosation* rate. The experimental error for each experimental point was estimated from the standard deviation of the spectral noise for one experiment.

**Figure 5 antioxidants-11-01236-f005:**
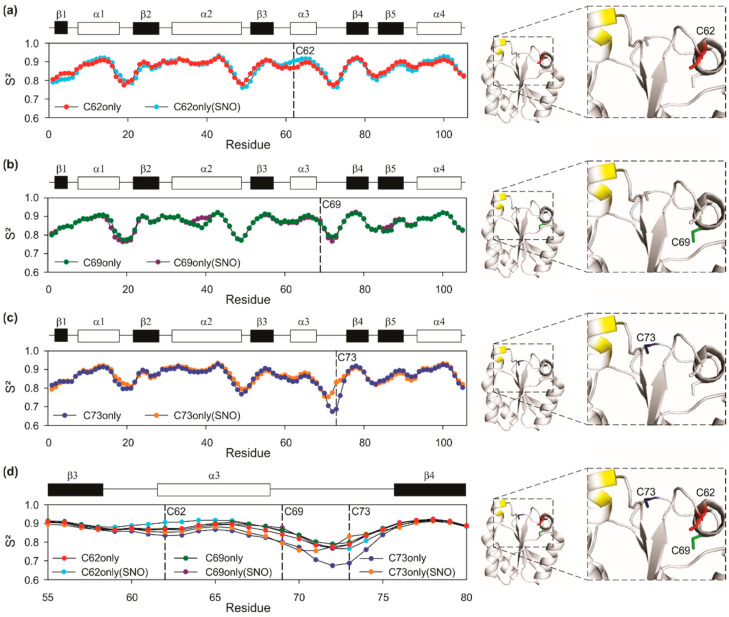
Random Coil Index for chemical shifts of the backbone. Each point on the graph represents a value for the order parameter (S^2^) calculated for each residue based on the Cα, CO, Cβ, N, Hα, and NH chemical shifts, the lower the S^2^ the greater the flexibility. On the right side, hTrx structures (PDB 4TRX). C62, C69, and C73 are colored in red, green, and blue, respectively. C32 and C35 are colored in yellow. (**a**) Red dots belong to the C62only mutant, and the cyan dots belong to the C62onlySNO mutant. From the *S-nitrosation* of residue C62, the region comprising residue S73 became more flexible, given a decrease in S^2^. The α3 helix, on the other hand, showed greater rigidity when residue C62 was S-nitrosated. (**b**) Green and magenta dots represent C69only and C69onlySNO residues, respectively. C69 *S-nitrosation* resulted in a small increase in flexibility of α3 and α3/β4 loop. (**c**) Blue and orange dots represent C73only and C73onlySNO residues, respectively. Note that C73 *S-nitrosation* resulted in a remarkable rigidity increase at α3/β4 loop, which characterizes an entropically unfavorable process. (**d**) Residues from the nitrosative site region with all mutants S^2^ plots combined.

**Figure 6 antioxidants-11-01236-f006:**
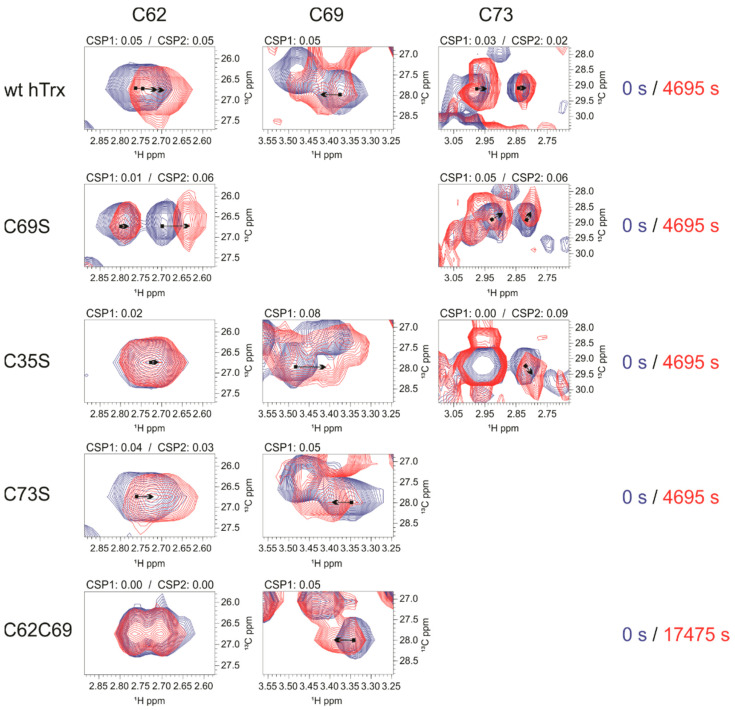
^1^H_β_-^13^C_β_ CSP of C62, C69, and C73 residues to hTrx and varied mutants. Blue ^1^H-^13^C HSQC corresponds to time zero, before GSNO addition, and the red one represents a reaction time after GSNO addition. The data show remarkably CSP to C62 to hTrx, C69S and C73S mutants, and a small CSP to C35S mutant. C62 residue did not change to C62C69. The results suggest that oxidation of the redox site affects the residue C62, *S-nitrosation* of C73 is less important to C62 CSP and C69 *S-nitrosation* does not affect C62. Concerning C69, all mutants and hTrx showed a noteworthy CSP, that is not related to either oxidation of the redox site or C73 *S-nitrosation*, but C62 *S-nitrosation*. In the same way, C73 is affected by C62 *S-nitrosation*.

**Figure 7 antioxidants-11-01236-f007:**
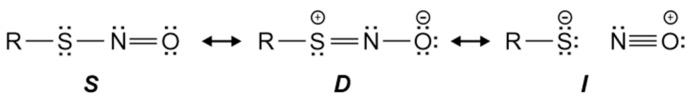
Resonance structures of S-nitroso group according Thalipov and co-workers [60] computational calculations.

**Figure 8 antioxidants-11-01236-f008:**
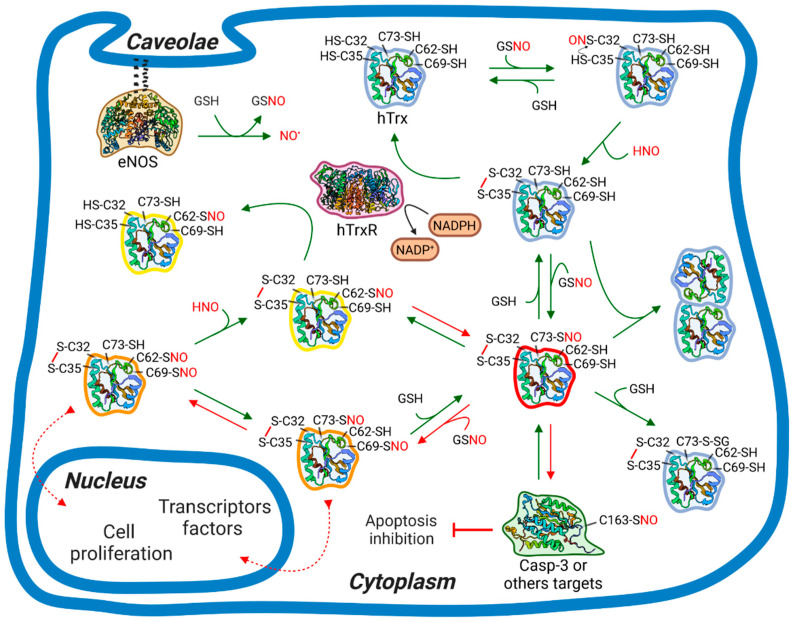
Hypothetical *S-nitrosation* path for hTrx based on the present findings and literature data: *S-nitrosation* of hTrx in a cell context. In the cell, eNOS produces NO that promptly reacts with GSH forming GSNO, responsible for transnitrosate C32 of reduced hTrx. Cys35 acts as a nucleophile and attacks C32-SNO to form a disulfide bond, leading to the oxidized form of hTrx. Reduced hTrx can be restored by NADPH-dependent hTrxR. After the oxidation of the hTrx redox site, the second step is the *S-nitrosation* of C73, the most solvent-exposed and reactive Cys residue. The *S-nitrosation* of C73 promotes the *S-nitrosation* of C62, the storage site, by a reversible NO internal transfer. Note that this mechanism regulates C73-SNO availability resulting in a control of Casp-3 *S-nitrosation* promoted by C73-SNO. Under nitrosative stress, GSNO also transnitrosate C69, a NO signaling pathway involved in hTrx translocation to the nucleus. C62 *S-nitrosation* exerts a cooperative effect in C69-SNO, leading to S-denitrosation of C69 residue and regulating the hTrx translocation to the nucleus. C73-SNO is also prone to thiolation reactions, such as hTrx dimerization and glutathionylation. The species are colored according to their reactivity, with blue as neutral or non-S-nitrosated, yellow, orange, and red with increasing *S-nitrosation* reactivity. Green arrows represent a physiological condition, whereas red arrows, nitrosative stress. hTrx, hTrxR, Casp-3 and eNOS are represented by PDB 4TRX [62], 2J3N [63], 1GFW [64], and 3NOS [65], respectively. Cys residue’s position is merely illustrative. Created with BioRender.com (accessed on 17 April 2022).

## Data Availability

All of the data is contained within the article and the Appendix A.

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
