# Peer review of "Deciphering the Path of S-nitrosation of Human Thioredoxin: Evidence of an Internal NO Transfer and Implication for the Cellular Responses to NO"

_antioxidants, 2022, doi:10.3390/antiox11071236_

Round 1
Reviewer 1 Report
The authors have generated a series of hTrx mutants to study the path of SNO and the relative contribution of each of the 5 Cys residues. They identified C62 as a slow, end-of-process NO storage site and C73 as the first being S-nitrosated and able to transfer NO intramolecularly to C62. NMR studies showed more pronounced impact of C62 nitrosation on hTrx structure than C69 or C73 nitrosation.
Specific Comments.
- While the studies are technically sound, the authors stopped short in documenting the biological relevance of their findings of Cys nitrosation. Figure 6 (on page 15) refers to “Casp-3 or others targets”. Where are these data to be found in the manuscript? Furthermore, there is another figure labeled as “Figure 6” on page 12.
- The Discussion often simply repeats the results section. Would it be possible to change the structure of the manuscript to avoid such unnecessary repetitions?
- Some figure (e.g. Figure 2a, b) appears to depict aggregate values with errors. However, the legends do not state n values, the type of error presented or whether the results were analyzed using statistical tests. The authors should also state the number of repetitions for other measurements (e.g. kinetics or concentration decay).
- Where are the data showing nuclear translocation of hTrx and regulation of proliferation (in what types of cells)?
- The text refers to Figure 7. Where is this figure to be found?
- Page 11. What does “C##only mutants” refer to?
- The title may be misleading for no cellular responses have been addressed in the mansucript.
Author Response
Reviewer 1
The authors have generated a series of hTrx mutants to study the path of SNO and the relative contribution of each of the 5 Cys residues. They identified C62 as a slow, end-of-process NO storage site and C73 as the first being S-nitrosated and able to transfer NO intramolecularly to C62. NMR studies showed more pronounced impact of C62 nitrosation on hTrx structure than C69 or C73 nitrosation.
Thank you for your positive evaluation.
- While the studies are technically sound, the authors stopped short in documenting the biological relevance of their findings of Cys nitrosation. Figure 6 (on page 15) refers to “Casp-3 or other targets”. Where are these data to be found in the manuscript? Furthermore, there is another figure labeled as “Figure 6” on page 12.
The present manuscript describes in detail with controlled experiments the path for the NO with the nitrosative site of hTrx. In the discussion section, we correlate our structural, dynamical, and biophysical findings with the biological data available in the literature and summarized it in Figure 7. To make it clearer in the revised version we added a title to this figure “Hypothetical scheme for the nitrosative role of hTrx based on the present findings and literature data”. The biological data of Casp-3 and other targets are vast in the literature and involve the participation of Cys73.
We also included in the discussion the following statement: “In the proposed mechanism we combined the data of the present work and the vast biological data available in the literature (Figure 7).”.
The Discussion often simply repeats the results section. Would it be possible to change the structure of the manuscript to avoid such unnecessary repetitions?
We carefully revised the discussion section to avoid repetition. In the revised manuscript the reference to data does not repeat the description of the data and the reference to literature is clearly stated. We thank the reviewer. We believe that the revised version was significantly improved.
Some figure (e.g. Figure 2a, b) appears to depict aggregate values with errors. However, the legends do not state n values, the type of error presented or whether the results were analyzed using statistical tests. The authors should also state the number of repetitions for other measurements (e.g. kinetics or concentration decay).
The experimental error analysis was conducted from the global fitting of multiple curves with different concentrations of GSNO. We used the software KinTek which uses confidence contour analyses, each of the fitting parameters is pushed to higher and lower values while allowing the other variables to be adjusted in deriving the best fit. A detailed description of the error analysis is described in reference 57. We added the following statement to the methods: “The experimental error analysis was conducted from the global fitting of multiple curves with different concentrations of GSNO. KinTek uses confidence contour analyses, each of the fitting parameters is pushed to higher and lower values while allowing the other variables to be adjusted in deriving the best fit [57].”
- Where are the data showing nuclear translocation of hTrx and regulation of proliferation (in what types of cells)?
The present manuscript describes in detail with controlled experiments the path for the NO with the nitrosative site of hTrx. In the discussion section, we correlate our structural, dynamical, and biophysical findings with the biological data available in the literature and summarized it in Figure 7. To make it clearer in the revised version we added a title to this figure “Hypothetical scheme for the nitrosative role of hTrx based on the present findings and literature data”. The biological data of hTrx nuclear translocation was described by Schroeder and cols in 2007 and it involves the S-nitrosation of Cys69 in a karyopherin-dependent manner.
- The text refers to Figure 7. Where is this figure to be found?
Corrected
- Page 11. What does “C##only mutants” refer to?
In the revised version we normalized the abbreviation for the cysteine only mutants Generically we named Cys_only mutants and for each of the mutants C32only, C35only, C62only, C69only, and C73only.
- The title may be misleading for no cellular responses have been addressed in the manuscript.
Thank you for this comment, the title was changed to “Deciphering the path of S-nitrosation of human thioredoxin: evidence of an internal NO transfer and implication for the cellular responses to NO” .
Reviewer 2 Report
The experimental design and data are satisfactory. Please improve writing and perform thiolate titrations of the single thiol mutants.

Author Response
Reviewer 2
The experimental design and data are satisfactory. Please improve writing and perform thiolate titrations of the single thiol mutants.
Thank you for the positive evaluation. We revised the writing and believe that the revised version was significantly improved.
We agree with the reviewer that the determination of the pKas of each of the cysteines in the Cys_only mutants would be informative. However, the present version of the manuscript is already too long. We thank the reviewer for the suggestion, and we will include this information in the near future work.
Reviewer 3 Report
Comments
- Title of the manuscript: the title goes beyond what the authors show with their data and therefore the title is too speculative. The title must be changed to “Deciphering the path of S-nitrosation in human thioredoxin: evidence of an internal NO transfer and implications for the cellular responses to NO”. these changes reflect that the authors only investigated S-nitrosation in thioredoxin and not S-nitrosation of other protein targets by S-nitrosated thioredoxin. Accordingly the authors only can suggest implications for the role of the S-nitrosation in thioredoxin for cellular responses to NO.
- The authors should cite and discuss their own related work in full detail (PMID: 33751377): “1 H, 15 N and 13 C backbone and side-chain assignments of reduced and S-nitrosated C62only mutant of human thioredoxin”.
- The authors should cite and discuss their own related work in full detail (PMID: 28808882): “1 H, 13 C and 15 N chemical shift assignments of Saccharomyces cerevisiae type 1 thioredoxin in the oxidized state by solution NMR spectroscopy”.
- In their methodds the authors should specify which Trx isoform they overexpressed and mutated – was it Trx1?
- Heading „2.2. Expression e purification of human Thioredoxin and mutants.“ Should be corrected – „e“ should be „and“?
- Figure 7. Please add title to the legend that this is a hypothetic scheme based on the present findings and literature data as you did not assess caspase trans-nitrosation of caspase or nuclear translocation of thioredoxin here. Also change the numbering in the legend to “Figure 7” as “Figure 6” is wrong here.
- In the last paragraph of the Discussion “In summary, we described…”, the authors should make clear that Figure 7 is a summary of own experimental data and literature data as you did not assess caspase trans-nitrosation of caspase or nuclear translocation of thioredoxin here.
Author Response
Reviewer 3
- Title of the manuscript: the title goes beyond what the authors show with their data and therefore the title is too speculative. The title must be changed to “Deciphering the path of S-nitrosation in human thioredoxin: evidence of an internal NO transfer and implications for the cellular responses to NO”. these changes reflect that the authors only investigated S-nitrosation in thioredoxin and not S-nitrosation of other protein targets by S-nitrosated thioredoxin. Accordingly, the authors only can suggest implications for the role of the S-nitrosation in thioredoxin for cellular responses to NO.
Thank you for the suggestion. We agree with the reviewer. The title was changed to “Deciphering the path of S-nitrosation of human thioredoxin: evidence of an internal NO transfer and implication for the cellular responses to NO”, as suggested.
- The authors should cite and discuss their own related work in full detail (PMID: 33751377): “1 H, 15 N and 13 C backbone and side-chain assignments of reduced and S-nitrosated C62only mutant of human thioredoxin”.
The manuscript was cited and discussed in the revised version.
- The authors should cite and discuss their own related work in full detail (PMID: 28808882): “1 H, 13 C and 15 N chemical shift assignments of Saccharomyces cerevisiae type 1 thioredoxin in the oxidized state by solution NMR spectroscopy”.
- In their methods the authors should specify which Trx isoform they overexpressed and mutated – was it Trx1?
We used Trx1. It was included in the Methods section.
- Heading „2.2. Expression e purification of human Thioredoxin and mutants.“ Should be corrected – „e“ should be „and“?
Corrected.
- Figure 7. Please add title to the legend that this is a hypothetic scheme based on the present findings and literature data as you did not assess caspase trans-nitrosation of caspase or nuclear translocation of thioredoxin here. Also change the numbering in the legend to “Figure 7” as “Figure 6” is wrong here.
We added the following title for Figure 6 legend: “Hypothetic scheme for the nitrosative role of hTrx based on the present findings and literature data:”
- In the last paragraph of the Discussion “In summary, we described…”, the authors should make clear that Figure 7 is a summary of own experimental data and literature data as you did not assess caspase trans-nitrosation of caspase or nuclear translocation of thioredoxin here.
Thank you for the suggestion. We added the following statement to the last paragraph: “The interpretation of the experimental data combined with the vast biological data available in the literature enabled us to propose a mechanism for the nitrosative cellular cycle.”.
Round 2
Reviewer 1 Report
The authors have adequately addressed most of my previous concerns. The revision has strengthened the manuscript. However, the manuscript will benefit from further editing and clarifications.
Page 2. Replace “cols” in “Barglow et cols” with “colleagues” or “co-workers”.
Page 3. “Thioredoxin” should be in lower case in the middle of the sentence.
Page 3. “their respectively” should read “their respective S-nitrosation…”
Page 3. “All these experiments” should read “results were analyzed…”
Page 4. “reaction to each mutant” should read “reaction for each mutant”
Page 5. Please check the heading of section 3.
Legend to Figure 2. Please indicate n values and variation (SD or SE?).
Figure 2 and Table 2 appear to present redundant information.
Legend to Figure 3. Please indicate whether the figure shows representative data of how many repeated measurements.
Page 8. The sentence beginning with “From now on we are going…” should be rephrased.
Heading of Section 3.3. Please remove “using” and replace “to describe” with “to define”.
Page 8. “we did not observe a significant difference” Insert “statistically” before “significant”.
Legend to Figure 4. Please indicate that these measurements were representative of n experiments. “These observations highlight C69 importance to..” should read “these observations highlight the importance of C69 to…”.
Legend to Figure 6. “In a similar fashion” would be better than “In the same way”.
Page 13. “changes in C73 are not due” should read “changes in C73 were not due”.
Page 14. “Moran and cols” should read “Moran and colleagues”.
Page 14. It is unclear what “These data impact the widely accepted interpretation” refer to. Do you mean “challenge”?
Page 15. “It is important to consider” should read “Of note…”
Page 15. “caspase-3” should be in lower case in the middle of the sentence.
Page 15. “We speculate that NO follows…” should read “we propose that…”.
Figure 7. The title is convulated, and the second half does not make sense. Please clarify “NOinternal transfer, a reversible mechanism proposed here”.
Page 16. “working as a stock” should be “functioning as a stock”. “in nitrosative stress” should read “under nitrosative stress”.
Page 16. “In opposition” should read “By contrast…”
Page 17. “stable and low reactive” should read “ stable with low reactivity”.
Page 17. “There are two main pitfalls in the strategy” should read “there are two major limitations to our strategy…”
Page 17, last sentence. “regulating cell responses and “ will be better.
Author Response
Reviewer 2
The authors have adequately addressed most of my previous concerns. The revision has strengthened the manuscript. However, the manuscript will benefit from further editing and clarifications.
Thank you for the positive evaluation and the careful and constructive review.
Page 2. Replace “cols” in “Barglow et cols” with “colleagues” or “co-workers”.
Changed. Also changed in two other parts of the manuscript.
Page 3. “Thioredoxin” should be in lower case in the middle of the sentence.
Corrected all along the manuscript.
Page 3. “their respectively” should read “their respective S-nitrosation…”
Corrected
Page 3. “All these experiments” should read “results were analyzed…”
Corrected
Page 4. “reaction to each mutant” should read “reaction for each mutant”
Corrected
Page 5. Please check the heading of section 3.
Checked. The comment was removed.
Legend to Figure 2. Please indicate n values and variation (SD or SE?).
We added the following text in the figure legend: “The experimental error analysis was conducted from the global fitting of multiple curves (n=1) with different concentrations of GSNO. The error bar refers to the standard deviation (SD).”.
Figure 2 and Table 2 appear to present redundant information.
The reviewer is correct. We removed Table 2.
Legend to Figure 3. Please indicate whether the figure shows representative data of how many repeated measurements.
Thank you for the comment. We added the following statement to all figures containing kinetic experiments: “The experimental error for each experimental point was estimated from the standard deviation of the spectral noise for one experiment.”.
Page 8. The sentence beginning with “From now on we are going…” should be rephrased.
We changed to “We will refer to …”.
Heading of Section 3.3. Please remove “using” and replace “to describe” with “to define”.
Changed
Page 8. “we did not observe a significant difference” Insert “statistically” before “significant”.
Changed
Legend to Figure 4. Please indicate that these measurements were representative of n experiments. “These observations highlight C69 importance to..” should read “these observations highlight the importance of C69 to…”.
Changed
We added the following statement to the legend: “The experimental error for each experimental point was estimated from the standard deviation of the spectral noise for one experiment.”.
Legend to Figure 6. “In a similar fashion” would be better than “In the same way”.
Removed
Page 13. “changes in C73 are not due” should read “changes in C73 were not due”.
Changed
Page 14. “Moran and cols” should read “Moran and colleagues”.
Changed
Page 14. It is unclear what “These data impact the widely accepted interpretation” refer to. Do you mean “challenge”?
To make it clearer, we change the statement to: “The similar S-nitrosation activity of wild type hTrx and C35S (mimics the reduced state) impacts the widely accepted interpretation that the oxidized and reduced hTrx have different S-transnitrosation activities [30], we suggested that the oxidized and reduced states have similar S-nitrosation activity, and the redox control may not be relevant.”.
Page 15. “It is important to consider” should read “Of note…”
Changed
Page 15. “caspase-3” should be in lower case in the middle of the sentence.
Changed
Page 15. “We speculate that NO follows…” should read “we propose that…”.
Changed
Figure 7. The title is convulated, and the second half does not make sense. Please clarify “NOinternal transfer, a reversible mechanism proposed here”.
We changed the title to: “Hypothetical S-nitrosation path for hTrx based on the present findings and literature data:”.
The statement referring to the NO internal transfer was changed to: “The S-nitrosation of C73 promotes the S-nitrosation of C62, the storage site, by a reversible NO internal transfer.”.
Page 16. “working as a stock” should be “functioning as a stock”. “in nitrosative stress” should read “under nitrosative stress”.
Changed.
Page 16. “In opposition” should read “By contrast…”
Changed
Page 17. “stable and low reactive” should read “ stable with low reactivity”.
Changed
Page 17. “There are two main pitfalls in the strategy” should read “there are two major limitations to our strategy…”
Changed
Page 17, last sentence. “regulating cell responses and “ will be better.
Changed.
Reviewer 2 Report
I have read the reply of the authors which is to the right direction. I have one last comment: the authors must clearly state somewhere in their manuscript (in the Discussion?) that measurement of the pKa values of the different thiols of human thioredoxin will further clarify the reactivity of cysteines with GSNO.
Author Response
We thank the reviewer for the positive evaluation. We added the following statement to the discussion: “The measurement of the pKa values of the different thiols could further clarify the reactivity of the cysteines to GSNO.”.